# A Computational Model of Hybrid Trunk-like Robots for Synergy Formation in Anticipation of Physical Interaction

**DOI:** 10.3390/biomimetics10010021

**Published:** 2025-01-02

**Authors:** Pietro Morasso

**Affiliations:** Robotic, Brain, and Cognitive Sciences Research Unit, Italian Institute of Technology, Center for Human Technologies, Via Enrico Melen 83, Bldg B, 16152 Genoa, Italy; pietro.morasso@iit.itt

**Keywords:** biomimetic robotics, cognitive robotics, soft robotics, hydrostat, neural simulation of action, prospection, passive motion, generative body schema, degrees of freedom problem

## Abstract

Trunk-like robots have attracted a lot of attention in the community of researchers interested in the general field of bio-inspired soft robotics, because trunk-like soft arms may offer high dexterity and adaptability very similar to elephants and potentially quite superior to traditional articulated manipulators. In view of the practical applications, the integration of a soft hydrostatic segment with a hard-articulated segment, i.e., a hybrid kinematic structure similar to the elephant’s body, is probably the best design framework. It is proposed that this integration should occur at the conceptual/cognitive level before being implemented in specific soft technologies, including the related control paradigms. The proposed modeling approach is based on the passive motion paradigm (PMP), originally conceived for addressing the degrees of freedom problem of highly redundant, articulated structures. It is shown that this approach can be naturally extended from highly redundant to hyper-redundant structures, including hybrid structures that include a hard and a soft component. The PMP model is force-based, not motion-based, and it is characterized by two main computational modules: the Jacobian matrix of the hybrid kinematic chain and a compliance matrix that maps generalized force fields into coordinated gestures of the whole-body model. It is shown how the modulation of the compliance matrix can be used for the synergy formation process, which coordinates the hyper-redundant nature of the hybrid body model and, at the same time, for the preparation of the trunk tip in view of a stable physical interaction of the body with the environment, in agreement with the general impedance–control concept.

## 1. Introduction

Two main “technological families” emerged during evolution to provide animal species with a high degree of motoric and manipulatory skills: muscle-activated skeletal systems and muscular hydrostats. In the case of the elephant, the two technologies appear to be functionally integrated as a unitary cyber-physical system that enhances the specific flexibility aspects of each motoric technology. However, if we compare the approximately 40,000 muscles of the elephant’s trunk, arranged in a combination of longitudinal, radial, transversal, and oblique muscle groups, with the relative paucity of muscles of the human body (650–840 skeletal muscles with 300–400 mono/bi/tri-articular joints), we may be induced to conclude that the two technological families are qualitatively different and that the hope to integrate them computationally for the design of soft or hybrid robotic systems in a coherent engineering framework might be just a dream, like artificial general intelligence.

Generally speaking, the conception and development of intelligent, autonomous and cooperative robots implies combining and merging a number of technologies: material technologies, sensor and actuator technologies, control technologies, and cognitive methodologies. The default conceptual framework assumes a kind of linear hierarchical arrangement of such technological items, with cognitive issues coming last and being strongly dependent on the previous design choices. The consequence is that, particularly in the case of robot designs inspired by biological muscular hydrostats, most research is focused on the initial stages of the hierarchy, which are characterized by a large variety of incompatible solutions, with little attention paid to the top functional level. In contrast, we suggest an alternative approach that reverses the hierarchy by focusing preliminarily on the general problem of synergy formation of hyper-redundant trunk-like robotic manipulators, independent of the specific implementation with a variety of materials, sensing and actuator technologies as well as the employed control strategies.

Moreover, we should not ignore the fact that, according to the theory of the neural simulation of actions [1], control technologies only apply to real (or overt) movements, whereas the more general synergy formation process, emerging from a cognitive framework, refers to both overt and covert (imagined) movements that are the basis of prospective capabilities and goal-oriented, adaptive behavior, both in humans and in autonomous robotics [2,3,4]. Along the same lines, it is worth considering that the general cognitive aspects of synergy formation for hyper-redundant kinematic structures may backwards influence the choice and configuration of the other technological, implementation aspects.

Elephant intelligence is well established from the behavioral point of view [5,6] and is supported by the fact that elephants have the largest brain of any land mammal [7]. Moreover, there is evidence that an elephant’s complex behaviors are produced by the combination of motion primitives and by computational mechanisms that reduce the biomechanical complexity of its body, including the trunk. In particular, the biomechanical analysis of natural movements [8] has demonstrated the presence of the following features: (a) reaching and fetching actions of the trunk are obtained by propagating inward curvature from the trunk tip; (b) the kinematics of the trunk tip is characterized by kinematic–figural constraints similar to the human arm gestures; and (c) the trunk can form semi-rigid segments connected by pseudo-joints during reaching movements. The working hypothesis, already considered in a previous paper on serpentine robots [9], is that the general features of elephant motion are just an extension of the features that characterize biological motion in humans [10,11], for both muscle-actuated overt (real) actions and mentally driven covert actions. In both cases, the theory of the neural simulation of action [1] suggests that the figural–kinematic invariants are already coded in the internal simulation model, independent of control strategies and muscle activation patterns.

A computational formulation of this approach is known as the passive motion paradigm (PMP) [12,13] and was recently extended from modeling the human body to hybrid robotic configurations that combine a skeletal-based component and a trunk-like component [9]. The idea is that the combination of the two components can be represented with a common format, i.e., a hyper-redundant kinematic chain characterized by a common Jacobian matrix *J*: such a matrix, together with a compliance matrix C, is the key element of the Generative body schema, which can be simulated, with the drive of primitive force field generators, producing coordinated movements of the hybrid hyper-redundant kinematic structure. In a previous study, we demonstrated that such a simulation model can reproduce the figural–kinematic invariants characteristic of natural elephant movements. In this paper, it is shown that the simulation model is also able to produce the inward wave of curvature of the trunk tip observed during reaching actions. Moreover, it is also shown that the appropriate modulation of the compliance matrix of the body schema can determine two relevant effects: (a) to form semi-rigid segments of the trunk during reaching; and (b) to implement the shape and orientation of the compliance ellipsoid of the trunk tip after reaching a target object, in preparation for appropriate interaction patterns.

We discuss our results in the context of so-called soft robotics, an emerging paradigm with the ambition of developing robotic manipulators replicating the high compliance, flexibility, and strength of natural hydrostats. In any case, it is worth stressing that the proposed computational model is not an alternative to the specific control technologies typically used in industrial robotics: it is just a general method for the coordination of a very large number of DoFs that is intrinsically consistent with the smoothness of biological motion. The choice and integration of control technologies refer to an implementation level, downstream of the design process, in relation to the stability and accuracy of the executed actions. Moreover, the same computational model has another fundamental function from the cognitive point of view, namely the simulation of imagined actions for evaluating the pros and cons of a given action before executing it. As regards the bio-inspired nature of the proposed synergy formation model, its uniqueness is the combination of two different, well-established streams of research in this area, namely the neural simulation of action [1] and the equilibrium-point hypothesis [14], thus establishing a computational link between motor control and motor cognition.

## 2. Methods

The implementation of the PMP model used in this simulation study is illustrated in Figure 1. The kinematic representation of the trunk is inspired by the piecewise constant strain (PCS) model or discrete Cosserat approach [15] derived from the Cosserat rod theory [16,17]: it represents the deformation of a trunk-like hydrostat as a series of micro solid rods that allow four modes of deformation (bending, twisting, stretching, and shearing). In the implementation reported in this paper, the model is limited to bending, but the extension to twisting and stretching is relatively straightforward. As suggested by Renda et al. [15], the computational models derived from the discrete Cosserat approach provide a unified mathematical framework linking traditional, skeletal-based robotics to soft robotics. Moreover, as will be further explained, this unification extends to hybrid robotic structures that include a skeletal and a hydrostatic segment. The crucial element of the PMP model is the Jacobian matrix of the hybrid elephant-like robot: it has between two and six rows (the Jacobian matrix of a planar kinematic chain has only two rows because the end-effector is identified by its position in the sagittal plane, while an additional row is required if the end-effector is characterized also by its pose; the full Jacobian, with six rows, corresponds to a spatial chain with an end-effector identified by both position and pose in 3D space) and a very large number of columns, virtually tending to infinity. In the majority of point-to-point reaching/transport tasks, the trunk displacement mainly occurs in the sagittal plane; thus, in this study, for simplicity, we limited to two the number of rows of the Jacobian matrix. The number of columns of J, corresponding to the skeletal part of the robot, is equal to the number of degrees of freedom (DoFs) of the skeletal part: in the present implementation, this number is six. The remaining columns of J identify the large number of functional units of the hydrostatic segment: in this study, this number is 48, with a total number of 54 DoFs. The Appendix A shows the algorithm used in the simulations for the computation of the Jacobian matrix, together with the related geometric parameters.

In the overall kinematic chain, the Jacobian matrix allows us to map an incremental variation δq of the joint rotation vector into the corresponding variation of the position/orientation vector of the end-effector δp: δp=J δq. This mapping expresses the well-posed direct kinematic transformation of the kinematic chain, whatever its degree of redundancy. Inverting this transformation is generally a difficult, ill-posed problem [18], particularly in the case of hyper-redundant robots [19,20]: unlike forward kinematics, inverse kinematics cannot be solved in a closed-form expression, providing a unique solution. Moreover, straight inverse kinematic methods cannot be easily extended to integrate complementary computational tasks, in addition to the pure kinematic task, as the satisfaction of the joint limits and the tasks related to the physical interaction with the manipulated objects is based on the modulation of the compliance of the end-effector. In particular, joint limits avoidance is a classical and crucial issue in robot control whose solution is typically based on the exploitation of kinematic redundancy, using optimization or iterative algorithms [21,22] whose complexity grows with redundancy, with a corresponding decrease in robustness.

The PMP approach bypasses the pure inverse kinematic goal by taking advantage of the double role of the Jacobian matrix [23]: in the mapping from the high-dimensional joint space to the low-dimensional end-effector space, the Jacobian matrix transforms incremental motions or the corresponding time derivative from the joint space to the end-effector space: ${\dot{p}}_{e}=J\;\dot{q}$; in the opposite direction, the Jacobian matrix maps a force vector $F_{e}$ applied to the end-effector onto the corresponding torque vector or focal torque field applied to the joints: $\tau_{foc}=J^{T}\;F_{e}$. This is also a well-posed mapping, whatever the degree of redundancy of the kinematic chain. The crucial point of the PMP is that it is force-based, not position-based; thus, the kinematic goal (a target $P_{T}$ to be reached by the end-effector) can be substituted with an elastic force field applied to the end-effector: $F_{e}=K\;\left(p_{T}-p_{e}\right)$. Moreover, in agreement with the experimental evidence from human biomechanics [24,25], the target point is actually a moving target ($p_{T}(t)$), shifting smoothly from the initial position of the end-effector ($p_{e}\left(t_{0}\right)$) to the final target ($P_{T}$) along a straight line and according to a prescribed duration (T). Thus, as shown in Figure 1, the synergy formation process that coordinates the hyper-redundant DoFs with the purpose of reaching a given target consists of integrating the following equation:(1)$$p_{e}\left(t\right)=\int_{0}^{T}JCJ^{T}K\left(p_{T}-p_{e}\right)dt$$

Two attractive force fields are implied by such an equation: a field that attracts the moving target to the final target and a field that attracts the end-effector to the current position of the moving target.

Figure 1 shows that a third force field contributes to the dynamics of the synergy formation process: the RoM protection field $\tau_{RoM}$ defined in the high-dimensional joint space. This is a torque field, with the function of protecting the motion of the kinematic chain from violations of the range of motion (RoM) of each DoF. It is a repulsive field [9] with a negligible amplitude for any joint of the kinematic chain, if the joint operates near the center of its RoM, and diverges exponentially when approaching the joint limits from either direction (repulsive torque field for RoM protection: $\tau_{RoM}(q)=k_{RoM}\left(e^{\frac{\left(q-q_{max}\right)}{\Delta }}-e^{\frac{{(q}_{min}-q)}{\Delta }}\right)$; the joint limits are $q_{min}$ and $q_{max}$). In other words, this simple RoM protection module is an alternative to the complex algorithms developed in industrial robotics for avoiding joint limits in the framework of inverse kinematics [21,22], whose robustness decreases with redundancy. Moreover, this module operates in real time, without the need for iterative optimization procedures, exploiting the additivity of the force/torque fields: it only needs the tuning of the repulsive parameters Δ and $k_{RoM}$.

The figure clarifies the force-based nature of the PMP model by making explicit the additivity of the force fields: the attractive focal field $\tau_{FoC}$ and the repulsive field $\tau_{RoM}$. The two fields operate simultaneously on the hyper-redundant kinematic chain and thus the complex inverse kinematic problem is transformed in a dynamic relaxation process to equilibrium. At the same time, it is worth considering that the force/torque fields mentioned above do not refer to physical entities and physical interactions but represent the virtual dynamics of the internal model that runs the synergy formation process for both overt and covert actions.

Let us now consider the multiple roles of the compliance matrix C, namely the modulation of the synergy formation process while it is enriched with additional subtasks, within the same computational structure related to Equation (1). In the implementation used for this simulation study, *C* is a diagonal 54 × 54 matrix, which can then be represented by a compliance vector, normalized by setting 1 as the highest value.

The main function of the compliance vector is to measure the degree of participation of each joint in the relaxation process that implements a given coordinated movement. For example, all the values of the vector can be set to 1 if we intend to maximize the flexibility of the hybrid kinematic chain, trunk included. In contrast, if we wish to freeze some part of the kinematic chain during a given action, in order to meet specific environmental constraints, it is sufficient to set to 0 (or to a very small value) the relevant DoFs, either of the skeletal or the hydrostatic segments. A variation/extension of this strategy is related to the observed capability of the elephant trunk [8] to form semi-rigid segments connected by pseudo-joints: this strategy can be achieved by setting to 0 (or very small values) all the DoFs except the DoFs that are supposed to emulate the pseudo-joints.

Another issue related to the modulation of the compliance matrix is related to the shape and orientation of the compliance ellipse of the end-effector at the end of a reaching movement, in anticipation of the interaction with a manipulated object, e.g., for pushing, hitting, etc. In the context of the PMP strategy, such an ellipse is determined by the eigen values and eigen vectors of the following matrix, which is an integral element of the PMP computational model, as shown in Figure 1:(2)$$C_{e}=I\;C\;J^{T}$$

In this simulation study, $C_{e}$ is a 2 × 2 matrix and C is a diagonal 54 × 54 matrix. The eigen values and eigen vectors of $C_{e}$ are a function of both the joint compliance vector (the main diagonal of matrix *C*) and the geometry of the kinematic chain (via the corresponding Jacobian matrix). Two parameters of the compliance ellipse are relevant from the interaction point of view: the degree of roundness (that depends on the ratio between the larger and the smaller eigen values) and the orientation (that depends on the eigen vector corresponding to the larger eigen values). The modulation of the compliance vector, in order to implement a desired compliance ellipse, is useful for the preparation of the kinematic chain for physical contact and haptic interaction with the environment at the end of a reaching movement.

One possibility is to implement a round ellipse. A perfectly round compliance ellipse, characterized by equal eigen values, implies an isotropic elastic interaction of the trunk tip with the touched object. This means that the force disturbances and displacement vectors are collinear in any direction, simulating a linear elastic behavior of the end-effector: this is desirable for the predictability and stability of the interaction. In contrast, isotropy is lost if the two eigen values are different. The computation of the joint compliance vector, which provides an approximately round compliance ellipse of the end-effector, may be carried out by means of an optimization procedure for the roundness indicator of the joint compliance vector (i.e., the ratio between the larger and the smaller eigenvalue of $C_{e}$): in the simulations, it was implemented by a simple gradient descent algorithm, starting from an initial setting to 1 of all the elements of the compliance vector.

Another option is to implement an ellipse with a desired orientation in relation to some feature or physical constraint of the environment. In this case, the computation of the desired ellipse orientation can be formulated in a similar way to the previously considered method to form semi-rigid segments of the trunk. Given the desired ellipse orientation vector, the method consists of tracing a straight line from the trunk tip perpendicular to the orientation line: the trunk segment intersected by that line can be chosen as the pseudo-joint of the trunk during the interaction with the manipulated object.

The modulation of the compliance vector and the associated force fields is the computational kernel of the PMP model, and it is largely independent of applications and implementation technologies. For specific designs of hybrid robots, the PMP model will need to be integrated with the sensory-motor control processes made available by the selected technologies.

From a computational point of view, the simulation model is an explicit system of first-order differential equations of high dimensionality. The simulations illustrated in the results was carried out using MATLAB^®^ (MathWorks, MATLAB R2023b), adopting the forward Euler method or the 4th-order Runge–Kutta method for integrating the differential equation system, with a time step of 0.1 ms. The simulations illustrated in the next section refer to a planar skeleton with 6 DoFs and a planar trunk with 48 DoFs. The simulation software is available on demand.

## 3. Results

Figure 2 shows the results of two equivalent simulations of a point-to-point upward-moving reaching gesture that share the same initial posture and final position of the trunk tip but slightly differ in the distribution of the compliance values. In both cases, as expected, the trajectory of the trunk tip is almost the same and approximately straight, with a bell-shaped profile.

These simulations are meant to illustrate the unitary coordination of the skeletal and hydrostatic segments, mediated by the compliance vector of the PMP model. All the DoFs participate in the motion with a different degree of involvement, quantified by the corresponding normalized compliance values: a zero value corresponds to a “frozen” DoF and a value of one means that the DoF is maximally responsive to the torque generated by the PMP model. The compliance vector remains constant over the whole movement: in both simulations, the compliance of the trunk is uniform and characterized by the maximum value of one, thus enhancing the flexibility of the trunk. In the simulation displayed in the left part of Figure 2, the compliance values of the six skeletal DoFs are small and linearly growing from the foot to the shoulder. In contrast, the simulation displayed in the right-side panels freezes the six DoFs of the skeleton: the skeleton remains fixed during the gesture and the trunk tip reaches the target with the same spatio-temporal pattern. However, the generated evolution of the shape of the trunk is quite different: smoother in the former case and more wrapped in the latter.

Thus, it appears that although the hyper-redundant nature of the trunk allows us to match the constraints involved in the given reaching gesture, independent of the recruitment of the skeletal part, a small amount of compliance provided to the skeletal DoFs is useful for avoiding the excessive stress of the trunk. In any case, the choice of the pattern of skeletal compliance is not critical; for example, a flat pattern with the same average value or the same linearly growing pattern with a double intensity would provide a qualitatively similar behavior. The choice between the two activation patterns of the compliance vectors in Figure 2 might be dictated by the anticipated constraints related to the next gesture that follows the final posture of the given gesture.

Figure 3 completes the analysis of the two gestures depicted in Figure 2: the top pair of graphs show the evolution of the 48 DoFs of the trunk during the two reaching movements, including the corresponding joint limits, illustrating that the RoM protection module of Figure 1 succeeded in maintaining each DoF inside the RoM interval; the bottom pair of graphs display the corresponding evolution of the curvature profile of the trunk from the initial instant $t_{0}$ to the final one $t_{f}$. Altogether, Figure 3 clarifies that the motion of the hyper-redundant trunk, during a coordinated global reaching movement, is approximately equivalent to a wave of inward propagation of the curvature from the trunk tip to the trunk base. This pattern, which smoothly coordinates the redundant DoFs of the trunk, is not specified explicitly but is the consequence of the PMP model that is primarily concerned with the motion of the end-effector, namely the trunk tip.

Figure 4 is focused on another crucial aspect associated with the modulation of the compliance vector, namely the simulation of pseudo-joints during the coordination of the hyper-redundant DoFs of the trunk in order to transform the hydrostat into a virtually articulated kinematic structure with a desired number of pseudo-joints and a desired length of the corresponding links. The left part of the figure illustrates the configuration of a two-DoF pseudo-articulated hydrostat and the right part the analogous configuration of a three-DoF structure. As shown in the graphs in the second row of the figure, this result is implicitly obtained by setting to one the hydrostat modules that correspond to the desired pseudo-joints and by freezing all the other modules, i.e., setting to a very low level the elements of the compliance vector. However, we should consider that, for a given trajectory of the trunk tip, the required rotation of the pseudo-joints might exceed the allowed RoM, in spite of the fact that the RoM protection component of the PMP model operates in order to avoid this danger. An additional option, to further reduce this danger, is to extend the “width” of the pseudo-joints, for example, from one to three elements, as illustrated in the graphs in the second row of the figure. The graphs in the third row show that this result is obtained by constraining all the DoFs inside the RoM. It is also worth comparing these graphs with the graphs in the middle row of Figure 3, which correspond to the same motion of the end-effector but with a totally compliant hydrostatic segment. This demonstrates that, although the choice of modulating the distribution of values in the compliance vector in order to simulate two or three pseudo-joints of the trunk increases the risk of overcoming the RoM of the hydrostatic structure, the extension of the width of the pseudo-joints, together with the RoM protection module, provides a simple and robust computational solution.

Figure 5 refers to the problem of modulating, at the end of a gesture, the compliance vector of the body model in such a way as to implement specific features of the compliance ellipse. Such features, e.g., the degree of roundness or the orientation, may be required for the stability or the accuracy of the interaction of the trunk tip with the environment or a manipulated object. The stiffness ellipse characterizes the geometrical aspects of the interaction, namely the relation between incremental relative motions and interaction force vectors. In the case of human movements, achieving a desired compliance ellipse is equivalent to selecting an appropriate pattern of coactivation of all the DoFs, on top of a distribution of activities that maintains the same equilibrium point [25,26]. It is suggested that a similar mechanism may apply to hybrid trunk-like robots. In the implemented version of the PMP model, the compliance vector is normalized, thus producing a compliance ellipse of a normalized size. The size of the ellipse, without affecting its shape, can be changed by applying a gain to all the elements of the compliance vector. The shape of the ellipse is selected as explained in the methods.

In particular, the left part of Figure 5 displays the activation of the compliance vector for achieving a round ellipse, i.e., an isotropic elastic behavior, according to which the elastic resistance force is always collinear, with a corresponding positional displacement and with the same amplitude, whatever the direction of the disturbance. The bottom graph in the left part of the figure shows that this solution is obtained with a coordinated modulation of the compliance of all the DoFs of both segments of the kinematic chain, namely the skeletal and the hydrostatic parts. In particular, the pattern of modulation of the hydrostatic part, which is characterized by a well-marked peak approximately in the middle of the trunk, reminds us of the pattern of modulation typical of pseudo-skeletal articulation of the hydrostat (illustrated in Figure 4), a single pseudo-joint in this case. It is also worth noting that such a peak of activity in the hydrostatic segment is associated with a peak of activity of the initial joints of the skeletal segment, enhancing the cooperation of the two parts of the body schema implicitly carried out by the PMP model. As explained in the methods, the modulation of the compliance vector capable of inducing a quasi-round compliance ellipse is obtained by a simple gradient-based descent process that minimizes the roundness indicator of the compliance ellipse: a few hundred steps were necessary to obtain the result in the figure. In general, the optimal compliance pattern for a quasi-round compliance ellipse is a function of the posture at contact. A sub-optimal strategy for avoiding a real-time computation immediately after the end of a reaching movement may be based on a memorization of the compliance patterns for frequently experienced posture and a related interpolation for the posture in play.

The right part of Figure 5 focuses on the computation of the compliance vector characterized by the fact that the orientation of the compliance ellipse approximately matches a desired angle, in agreement with the shape and physical properties of the manipulated object. In this particular example, the desired orientation is 45 deg: the solution easily found in this case is the identification of a pseudo-joint in the trunk, such that small incremental rotations around the joint displace the end-effector along the desired angle. The pseudo-joint is computed by tracing a line from the end-effector, perpendicular to the required orientation of the ellipse, and selecting the DoF of the kinematic chain that is mostly collinear with the traced line. The selected DoF is assigned a maximum compliance value and all the other DoFs are frozen, as shown in the bottom graph in the right part of Figure 5. This is equivalent to tuning the compliance vector for simulating the best one-joint segmentation of the kinematic chain. If no DoF is sufficiently aligned with the traced line, it is possible to adopt a minimization strategy similar to the one applied to achieve a round ellipse. For example, it is possible to identify the pair of DoFs that are best aligned with the specified direction and then use the pair of compliance parameters to guide the gradient descent. According to this strategy, all the other joints should be frozen and this is equivalent to tuning the compliance vector for simulating the best two-joint segmentation of the kinematic chain. In any case, after having reached the target point using a specific assignment of the compliance vector, it is possible to immediately evaluate the orientation of the compliance ellipse for that posture by computing the eigenvector related to the higher eigenvalue of the compliance matrix (Equation (2)) and evaluate if that orientation matches, at least approximately, the expected interaction patterns.

At the same time, it is worth considering that the force/torque fields mentioned above do not refer to physical entities and physical interactions but represent the virtual dynamics of the internal model that runs the synergy formation process for both overt and covert actions.

## 4. Discussion and Conclusions

Soft robots are made from soft materials instead of hard ones, such as silicone, rubbers, and gels, structured in such a way as to allow control of the shape, similarly to living hydrostats such as octopus tentacles or elephant trunks [27,28]. Both types of hydrostats do not have bones but can stiffen up, when needed, for a variety of dexterous behaviors, such as to grab food or use tools [29]. The elephant’s trunk is a complex network composed of tens of thousands of bundles of muscle fibers, which work together by relaxing and contracting to produce forces that shape the trunk, driving the trunk tip to the target [30,31]. Although trunks do not have any bone or semi-hard cartilage, the motor dexterity of elephants depends on the close sensory–motor–cognitive integration between the soft hydrostatic part and the articulated skeletal part; in contrast, this requirement is irrelevant for the octopus because it is not a terrestrial animal. Thus, for a large part of the foreseeable applications that may involve soft robots, the choice should aim to hybridize systems that combine a soft and a hard subsystem, with a hyper-redundant kinematic capability. The motivation for conceiving soft or hybrid robots emerged recently from considerations of the limitations of traditional industrial robots in negotiating natural environments in the general framework of the bio-inspired evolution of service robotics with a high degree of human–robot interaction and an embodied cognitive approach [27,32].

In this study, it is shown how the PMP computational model is naturally compatible with hybrid robotic systems, integrating the soft and the hard components in a common computational framework, independent of the specific technology used for the soft part. As explained in the methods, such a computational framework is fundamentally force-based, and we may observe that for a large family of experimental soft robots, the design and control technologies are force-based as well. Consider, for example, a method called jamming [33,34,35], namely a process for enabling controllable stiffness and induced shape changes in soft robotics, in which granular materials in the trunk-like arm are packed together to change its stiffness, or the soft origami modular (SOM) segment concept [36]. In contrast, traditional rigid robots for industrial applications are mostly position-controlled.

We should also observe that the PMP approach is force-based for both the soft and rigid components, whatever the degree of redundancy of the robot kinematic schema, thus providing a natural integration framework for the two subsystems. Moreover, the PMP approach provides a rational and natural method for modulating the overall stiffness of the hybrid robot performance in a functional manner. For example, a compliant, flexible state is appropriate for delicate tasks and for assuring the safety of people interacting around the robot; on the other hand, a more rigid state for carefully selected segments of the kinematic chain is helpful when dealing with heavy loads and challenging tasks.

In general, a crucial role of stiffness is related to the difficulties engendered by contact tasks that require intimate dynamic interaction between the robot and its environment: that interaction changes the performance of the robot and can jeopardize the stability of its control system. Contact stability may be guaranteed if the control system provides the manipulator with an appropriately structured dynamic response to environmental inputs, characterized as impedance control [37,38].

Generally speaking, trunk-like soft arms may exhibit high dexterity and adaptability very similar to the elephants. However, owing to the continuum and soft bodies, their performance in payload and spatial movements is limited, although numerous preliminary results have been achieved in soft robotic applications, e.g., for manipulation [39,40] and grasping [41]. Moreover, the transition from laboratory prototypes to real-world applications requires a deeper understanding of the limits of the current approaches in terms of the materials, design principles and related control methodologies. Thus, we suggest that the computational approach investigated in this study is fundamentally independent of the specific technological choices, providing general principles of synergy formation and compliance modulation that may help consolidate the emergence of hybrid robot technologies.

As regards the efficiency of the proposed synergy formation process, we may consider that its intrinsic dynamics are structured around the imitation of the spatio-temporal features of biological motion, with a particular emphasis on smoothness as a minimization of jerk [42]. In general, the energetic frugality determined by smoothness has been demonstrated in human motor control [43], as well in industrial robotics [44]. Moreover, since the general smoothness is independent of the chosen modulation patterns of the compliance vector, we may expect energetic frugality to be invariant throughout compliance modulation. At the same time, we may expect that the energy consumption and motion accuracy of specific implementations and applications may depend on the employed hardware and control firmware, but we are confident that the integration with the proposed synergy formation model is likely to be beneficial.

As regards the cognitive framework of the synergy formation process, future developments should focus on its integration in a specific cognitive architecture. This research field has been very active for several decades and a recent review [45] reported tens if projects at different levels of development, but we are still far away from some sort of standard framework. Moreover, the family of cognitive architectures is divided between architectures that aim at modeling human cognition in general, as a unified theory of cognition, like CLARION [46], SOAR [47], ACT-R [48], and other architectures that focus on cognitive robotics, such as ISAC [49], ArmaX [50], and CRAM [51], in order to allow a robot to accomplish manipulation tasks typical of everyday life with the versatility and flexibility that are exhibited by humans as well as many mammals, including elephants. A cognitive architecture is a software framework that integrates a number of essential cognitive functions for an autonomous cognitive agent, such as active perception, purposive action, learning, adaptation, anticipation, prospection, motivation, attention, action selection, memory, reasoning. In particular, internal simulation, as a tool for synergy formation and prospection, is a crucial module that could be integrated into the hybrid symbolic/sub-symbolic organization of the robotic cognitive architectures above.

At the same time, we should consider that, in contrast to industrial robots that are programmed for well-defined tasks in controlled and predictable scenarios, autonomous robots for open scenarios do need a cognitive architecture that can operate with two alternative but complementary spatio-temporal modalities, namely the modality of covert action (e.g., the PMP model) vs. the modality of overt action, switching between them according to decision-making and knowledge of the results, thus accumulating experience and know-how using different learning procedures such as reinforcement learning. In particular, when entering the overt modality, the robot will recruit real-time processes, appropriate for the given task and compatible with the accomplished skill level, including movement guidance and various forms of haptic interaction such as tactile servoing or tactile exploration.

## Figures and Tables

**Figure 1 biomimetics-10-00021-f001:**
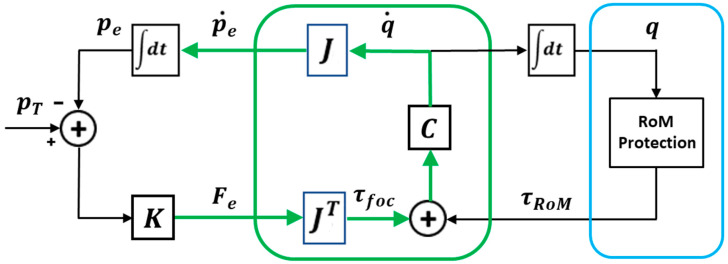
PMP computational model for the synergy formation and compliance modulation of a hybrid kinematic chain with a skeletal segment and a hydrostatic segment. The RoM protection module is an algorithm for avoiding joint limits.

**Figure 2 biomimetics-10-00021-f002:**
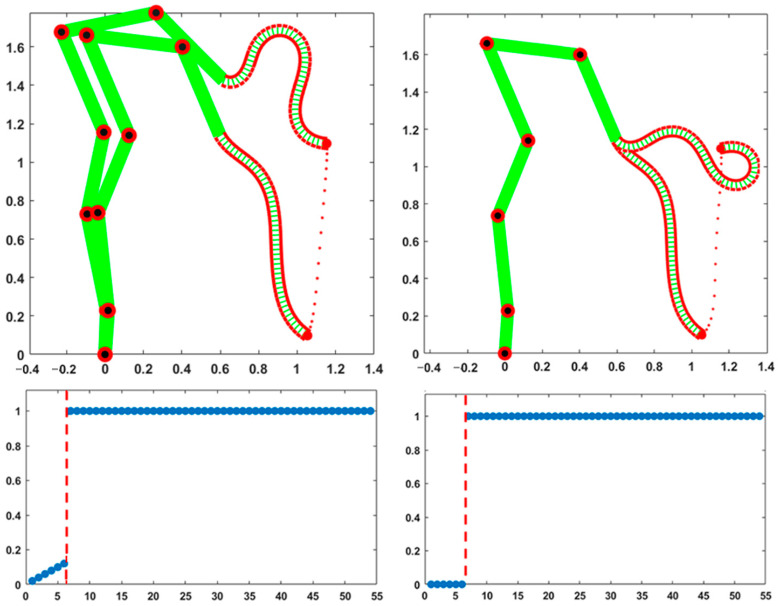
Two simulations of a point-to-point upward-moving reaching gesture (measurement units are meters for both axes), related to a slightly different distribution of the compliance vectors, illustrated in the two bottom panels: the 54 elements of the normalized vector are ordered, the skeletal part first (the first 6 elements from foot to shoulder), followed by the hydrostatic part (the following 48 elements from the trunk base to the trunk tip). The dashed red line separates the DoFs of the skeletal segment from the DoFs of the hydrostatic segment. In the two top panels, the same compliance values are visualized with a color code according to the grayscale palette.

**Figure 3 biomimetics-10-00021-f003:**
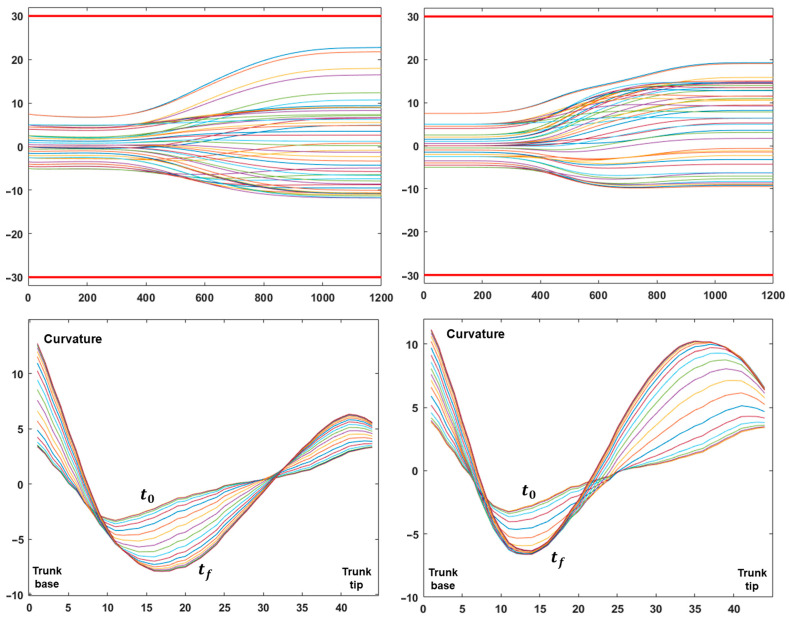
The left/right parts of the figure refer to the gesture of the left/right parts of Figure 2. For each column, the top graph shows the time course of the 48 DoFs of the trunk; the measurement units are ms for the time axis and deg, for the vertical axis; the two red horizontal lines identify the RoM of the trunk DoFs (±30 deg). The bottom graphs show the evolution of the trunk curvature profile (measurement unit m^−1^, from the trunk base to the trunk tip) during the corresponding gesture, from the initial to the final instant of the movement.

**Figure 4 biomimetics-10-00021-f004:**
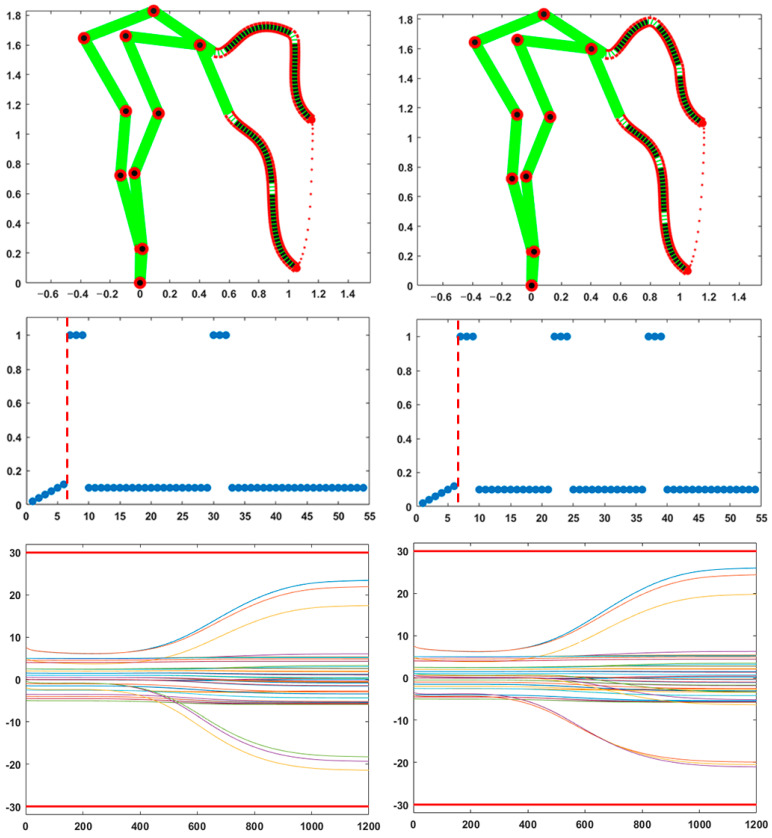
The two columns of graphs illustrate two simulations of a point-to-point upward-moving gesture with the same initial posture and final position of the trunk tip but with different compliance vectors of the trunk segment. In the left column, the vector of the trunk is set to simulate a skeletal articulation with 2 pseudo-joints and 3 rigid segments. In the left column, the skeletal articulation of the trunk has 3 pseudo-joints and 4 rigid segments. The top row of graphs shows the initial and final body postures and the trajectory of the trunk tip (measurement unit meters for both axes). The corresponding compliance vectors are plotted in the middle row of the graphs: 54 DoFs, ordered from foot to trunk-tip; the compliance vectors are also visualized in the top row of the graphs, with a color code according to the grayscale palette. The bottom row of graphs plots the time course of the 48 DoFs of the trunk (measurement unit ms for the time axis and deg for the vertical axis); the red horizontal lines identify the RoM of the trunk DoFs.

**Figure 5 biomimetics-10-00021-f005:**
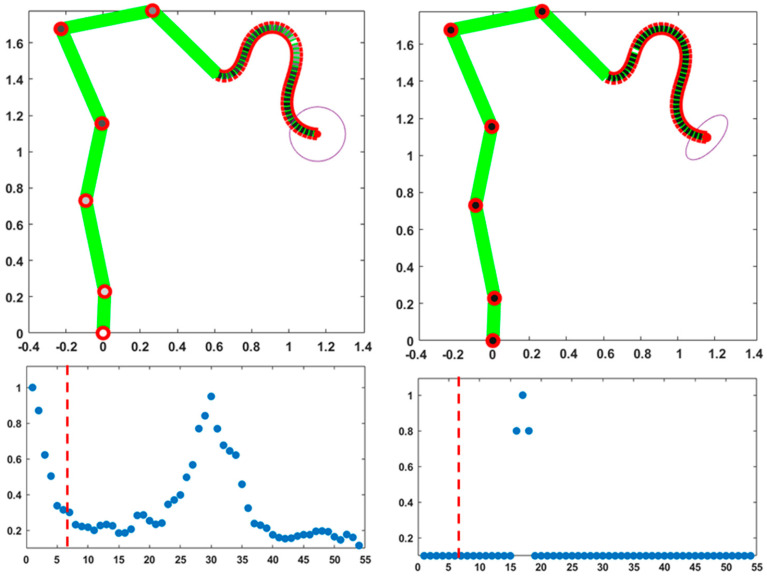
Modulation of the compliance vector for achieving a desired compliance ellipse of the end-effector, for a posture achieved at the end of a reaching movement. The left column of graphs refers to a desired round ellipse; the right column to an ellipse oriented according to a 45 deg angle. The corresponding compliance vectors are plotted in the bottom row of graphs: both graphs show the normalized compliance values of the 54 DoFs of the model, ordered from the foot to the trunk tip. The same compliance values are visualized in the top row of graphs, with a color code according to the grayscale palette. The dashed red line separates the DoFs of the skeletal segment from the DoFs of the hydrostatic segment. The stiffness ellipses corresponding to the plotted values of the compliance vector are visualized around the trunk tip.

## Data Availability

The simulation software is available on demand.

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
