# Peer review of "A Computational Model of Hybrid Trunk-like Robots for Synergy Formation in Anticipation of Physical Interaction"

_biomimetics, 2025, doi:10.3390/biomimetics10010021_

Round 1
Reviewer 1 Report
Comments and Suggestions for Authors
The manuscript presents a computational model for hybrid trunk-like robots, aiming to address the synergy formation and compliance modulation in anticipation of physical interaction. The work is in the context of soft and hybrid robotics, with an emphasis on integrating soft and hard components inspired by the elephant's body structure. Before publication, the author should address the following issues:
1. Although the article proposes a novel method, the comparison with other related studies is not thorough enough. It fails to fully explore the similarities and differences in methods, performance, and application scenarios between this model and other similar soft or hybrid robotic models. For example, when elaborating on its computational model, it does not clearly illustrate the unique innovations of this model in handling synergy motion and compliance modulation compared to traditional industrial robot control methods or other bio-inspired robotic models.
2. Despite the simulation study, the discussion on the feasibility, potential challenges, and coping strategies of the model in practical applications is relatively limited. In real scenarios, robots may face various complex environmental conditions, task requirements, and operational constraints. The article should analyze more deeply how the model performs under these practical circumstances. For instance, when the robot interacts with objects of different shapes, weights, and surface characteristics, how does the model ensure stable and effective operation? How does the robot's motion planning and synergy control adapt in complex workspaces such as narrow passages or environments with many obstacles? These issues are not fully addressed.
3. The explanations of some technical concepts and terms may not be easily understandable for readers who are not familiar with the professional knowledge in this field. For example, the description of the "RoM protection field" and its interaction mechanism with other force fields is rather abstract, and no more intuitive methods (such as diagrams, examples for analogy, etc.) are used to help readers better understand its role and impact in the entire model. In addition, the explanations of the meanings and physical implications of some mathematical formulas and matrix operations can also be strengthened to enable readers to follow the research ideas more easily.
4. The article does not deeply explore how to achieve such dynamic adjustment more intelligently and efficiently, and the impact of different modulation strategies on the robot's energy consumption, motion accuracy, and response speed.
5. The introduction emphasizes the importance of the cognitive framework for the synergy formation process, but in the subsequent model description and experimental analysis, the detailed explanation of how the cognitive level specifically affects the robot's synergy motion and decision-making process is not enough.
Author Response
Comment 1. Although the article proposes a novel method, the comparison with other related studies is not thorough enough. It fails to fully explore the similarities and differences in methods, performance, and application scenarios between this model and other similar soft or hybrid robotic models. For example, when elaborating on its computational model, it does not clearly illustrate the unique innovations of this model in handling synergy motion and compliance modulation compared to traditional industrial robot control methods or other bio-inspired robotic models.
Response 1. The proposed computational model is not an alternative to specific control technologies typically used in industrial robotics: it is just a general method for the coordination of a very large number of DoFs that is intrinsically consistent with the smoothness of biological motion. The choice and integration of control technologies refers to an implementation level, downstream the design process, in relation with the stability and accuracy of executed actions. This is the reason for which a systematic comparison with traditional industrial robotic control methods is not carried out, except for the issue of “joint limits avoidance” problem well known in industrial robotics in the framework of inverse kinematics for highly redundant robots (lines 158-164, 190-195). Moreover, the same computational model has another fundamental function from the cognitive point of view, namely the simulation of imagined actions for evaluating the pros and cons of a given action before executing it. As regards the bio-inspired nature of the proposed synergy formation model, its uniqueness is the combination of two different, well-established streams of research in this area, namely the neural simulation of action and the equilibrium-point hypothesis, thus establishing a computational link between motor control and motor cognition.
Comment 2. Despite the simulation study, the discussion on the feasibility, potential challenges, and coping strategies of the model in practical applications is relatively limited. In real scenarios, robots may face various complex environmental conditions, task requirements, and operational constraints. The article should analyze more deeply how the model performs under these practical circumstances. For instance, when the robot interacts with objects of different shapes, weights, and surface characteristics, how does the model ensure stable and effective operation? How does the robot's motion planning and synergy control adapt in complex workspaces such as narrow passages or environments with many obstacles? These issues are not fully addressed.
Response 2. We should consider that, in contrast with industrial robots that are programmed for well-defined tasks in controlled and predictable scenarios, autonomous robots for open scenarios do need a cognitive architecture that can operate with two alternative but complementary spatio-temporal modalities, namely the modality of covert action (e.g., the PMP model) vs. the modality of overt action, switching between them according to decision-making and knowledge of results, thus accumulating experience and know-how using different learning procedures such as reinforcement learning. In particular, when entering the overt modality, the robot will recruit real-time processes, appropriate for the given task and compatible with the accomplished skill level, including movement guidance and various forms of haptic interaction such as tactile servoing or tactile exploration.
Comment 3. The explanations of some technical concepts and terms may not be easily understandable for readers who are not familiar with the professional knowledge in this field. For example, the description of the "RoM protection field" and its interaction mechanism with other force fields is rather abstract, and no more intuitive methods (such as diagrams, examples for analogy, etc.) are used to help readers better understand its role and impact in the entire model. In addition, the explanations of the meanings and physical implications of some mathematical formulas and matrix operations can also be strengthened to enable readers to follow the research ideas more easily.
Response 3. I modified the text with more explanations (lines 159-165, 192-197) specifying that the “RoM protection module” corresponds to the “joint limits avoidance” problem well known in industrial robotics with uniquely distinguishing characteristics. The discussion on this issue is complemented with two new quotations (21 &22) and a new footnote.
Comment 4. The article does not deeply explore how to achieve such dynamic adjustment more intelligently and efficiently, and the impact of different modulation strategies on the robot's energy consumption, motion accuracy, and response speed.
Response 4. The issue of energy efficiency is discussed in the Discussion (lines 441-451) with three more citations (42, 43, 44).
Comment 5. The introduction emphasizes the importance of the cognitive framework for the synergy formation process, but in the subsequent model description and experimental analysis, the detailed explanation of how the cognitive level specifically affects the robot's synergy motion and decision-making process is not enough
Response 5. As regards the cognitive framework of the synergy formation process, it is discussed in the Discussion (lines 452-465) with additional citations (45-51).
Reviewer 2 Report
Comments and Suggestions for Authors
Reviewer Comments
1. Overall comments
The author proposed a simulation model based on Passive Motion Paradigm (PMP) to implement the synergy formation process and adjustment end effector orientation of hybrid trunk-like robots. The model calculates the hybrid kinetic chain using the Jacobian matrix and maps the force field into the body gestures using the compliance matrix. It was found that the modulation of the compliance matrix can achieve the same trajectory with different movements and desired end-effector orientation. The proposed model provides a common computation framework to integrate the hard and soft parts of hybrid robots, which is independent of specific technologies and can help understand the relationship between the compliance/stiffness of the robot body and its physical interaction with objects. This work is promising for accurate, robust and real-time control of hybrid robots. The manuscript is well-written in a coherent and logical flow, but a few minor revisions are required before publication. Please see the specific comments below.
2. Specific comments
1) There are a few typos in the manuscript, such as ‘pseud-joints’ in line 176 and ‘relatedd’ in line 253.
2) In line 289, it should be ‘the second row of figure 2’ instead of ‘the middle row of figure’ that the author intended to compare. Please check and correct it if needed.
3) In line 356, the desired orientation is 45 degrees but Figure 5 caption mentioned that it is 30 degrees. Please check and correct it if needed.
4) In line 345, it should be ‘simple gradient-based’ instead of ‘simple gradient based’. Please check and correct it if needed.
5) In line 443, the publication year of reference 9 should be 2024 instead of 2001. Please check and correct it.
6) If possible, can the author append the simulation model Matlab script for others to repeat this work?

Author Response
Specific comments
1) There are a few typos in the manuscript, such as pseud-joints in line 176 and related in line 253.
OK
2) In line 289, it should be the second row of figure 2 instead of the middle row of figure that the author intended to compare. Please check and correct it if needed.
OK
3) In line 356, the desired orientation is 45 degrees but Figure 5 caption mentioned that it is 30 degrees. Please check and correct it if needed.
OK
4) In line 345, it should be simple gradient-based instead of simple gradient based . Please check and correct it if needed.
OK
5) In line 443, the publication year of reference 9 should be 2024 instead of 2001. Please check and correct it.
OK
6) If possible, can the author append the simulation model Matlab script for others to repeat this work?
The Matlab script is available on demand (lines 265,266 of the methods)
Round 2
Reviewer 1 Report
Comments and Suggestions for Authors
The author has addressed my issues.